# Fusion of Spectral and Structural Information from Aerial Images for Improved Biomass Estimation

**Bikram Pratap Banerjee [1]** , **German Spangenberg [2,3]** and **Surya Kant [1,4,\*]**

1   Agriculture Victoria, Grains Innovation Park, 110 Natimuk Rd, Horsham, Victoria 3400, Australia;
    bikram.banerjee@agriculture.vic.gov.au
2   Agriculture Victoria, AgriBio, Centre for AgriBioscience, 5 Ring Road, Bundoora, Victoria 3083, Australia;
    german.spangenberg@agriculture.vic.gov.au
3   School of Applied Systems Biology, La Trobe University, Bundoora, Victoria 3083, Australia
4   Centre for Agricultural Innovation, School of Agriculture and Food, Faculty of Veterinary and Agricultural
    Sciences, The University of Melbourne, Victoria 3010, Australia
*   Correspondence: surya.kant@agriculture.vic.gov.au; Tel.: +61-3-4344-3179

**Abstract:** Efficient, precise and timely measurement of plant traits is important in the assessment of a breeding population. Estimating crop biomass in breeding trials using high-throughput technologies is difficult, as reproductive and senescence stages do not relate to reflectance spectra, and multiple growth stages occur concurrently in diverse genotypes. Additionally, vegetation indices (VIs) saturate at high canopy coverage, and vertical growth profiles are difficult to capture using VIs. A novel approach was implemented involving a fusion of complementary spectral and structural information, to calculate intermediate metrics such as crop height model (CHM), crop coverage (CC) and crop volume (CV), which were finally used to calculate dry (DW) and fresh (FW) weight of above-ground biomass in wheat. The intermediate metrics, CHM ($R^2$ = 0.81, SEE = 4.19 cm) and CC (OA = 99.2%, K = 0.98) were found to be accurate against equivalent ground truth measurements. The metrics CV and CV×VIs were used to develop an effective and accurate linear regression model relationship with DW ($R^2$ = 0.96 and SEE = 69.2 g/m$^2$) and FW ($R^2$ = 0.89 and SEE = 333.54 g/m$^2$). The implemented approach outperformed commonly used VIs for estimation of biomass at all growth stages in wheat. The achieved results strongly support the applicability of the proposed approach for high-throughput phenotyping of germplasm in wheat and other crop species.

**Keywords:** high-throughput phenotyping; wheat; plant breeding; crop coverage; crop volume

## 1. Introduction

Above-ground biomass (hereafter mentioned as biomass) of a crop is an important factor in the study of plant functional biology and growth, being the basis of vigour and net primary productivity [1–3]. The biomass is a measure of the total dry weight (DW) or fresh weight (FW) of organic matter per unit area at a given time [1,4]. As biomass is directly related to yield, particularly in later crop growth stages, it is crucial for monitoring biomass to estimate grain yield. For example, grain yield is determined using biomass in the AquaCrop model [5]. The traditional method to measure biomass is performed by destructively harvesting fresh plant material and weighing the samples for FW, and oven drying the samples to get DW. The process performed on large experiments is time-consuming and labour intensive. Therefore, a high-throughput imaging method using an unmanned aerial vehicle (UAV) has been proposed to infer crop biomass accurately as a non-destructive and fast alternative.

Near-earth remote sensing technologies from UAVs have evolved considerably for application in high-throughput field phenotyping for plant trait measurement [6]. Collecting multispectral imagery

from UAVs is simple and cost-effective [7], with UAVs helping to overcome limitations imposed by traditional in-field trait scoring, and the destructive harvesting of samples for biochemical assays. Crop canopy reflectance can be remotely sensed, providing information on the biochemical composition (e.g., chlorophyll content, leaf water content, fresh or dry matter content), canopy structural parameters (e.g., leaf-area index, leaf angle), and soil properties (e.g., soil optical properties, soil moisture) [8,9].

Several vegetation indices (VIs), such as normalized difference vegetation index (NDVI [10]), the optimised soil-adjusted vegetation index (OSAVI [11,12]), and the modified soil-adjusted vegetation index (MSAVI [13]), were developed utilizing the reflected visible and near-infrared radiation from the crop canopy to simulate biophysical and biochemical traits, including biomass [14–19]. Both physical models and empirical regression techniques are used to derive desired crop traits [20,21]. Physical models are derived using physical principles, such as the PROSAIL model [8,9]. However, different parameters for physical models are often not readily available, limiting the practical application in estimating significant crop parameters [22,23]. Regression techniques are data-driven approaches which are often used to relate spectral information with the desired traits, such as biomass [24,25]. Conventional regression techniques (multiple linear regression, multiple stepwise regression techniques, partial least squares regression) and machine learning (artificial neural networks, random forest regression, support vector machine regression) are also often used when the spectral information does not relate well with the desired trait. This can occur when suffering from low signal-to-noise ratios or there are additional factors that influence the spectral information.

Typically, spectral VIs, which are widely used, strongly correlate with crop parameters, including biomass, during the vegetative growth stages [21,26]. However, these traditional VIs lose their sensitivity in reproductive growth stages, thereby making it difficult to establish an estimation model that works over whole of the life cycle [21,27]. Furthermore, VIs computed using the red and near-infrared spectral bands tend to saturate at high canopy coverage [28,29]. Several studies have investigated different approaches for improving biomass estimation using (i) hyperspectral narrow-bands for novel VI development [15,16,30–34], (ii) additional information from light detection and ranging (LiDAR) [3,35–37], (iii) complementary information synthetic aperture radar (SAR) [38–41], (iv) crop structure calculations, such as surface models from UAVs [42–44], and (v) ultra-high-resolution image textures from UAVs [45].

In addition, the images from UAV surveys can be photogrammetrically restituted to produce a digital surface model (DSM) during the creation of a multispectral ortho map, from which crop height models (CHMs) can be developed [3,35,43,46]. Methods based on the use of UAV photogrammetry-derived CHM for estimating biomass were found to be effective in different field crops [47,48]. CHMs can provide complementary three-dimensional information, which opens new opportunities for deriving biomass in field crops [42,49,50]. Using both spectral (VIs) and structural (surface models) information obtained from UAVs could be a useful approach for not only estimating biomass but also other crop traits such as leaf area index, chlorophyll and nitrogen content, plant lodging, plant density, and counting head numbers. Though few studies have combined spectral (VIs) and structural (CHMs) data in estimating biomass in barley [43], pastures [51], and shrubs-dominated ecosystems [52]. However, the potential of combining spectral and structural information for the estimation of dry and fresh biomass in the agronomically important food crop wheat is yet to be investigated. Additionally, none of the previous studies successfully constructed a model using UAV-derived information that highly correlates with biomass during reproductive and senescence stages across breeding trials with diverse genotypes.

The objectives of this study are to evaluate the use of images collected from UAV multispectral surveys in generating complementary multispectral orthomosaic and canopy surface information, and subsequently fuse the two information layers to accurately model biomass in wheat. This study presents biomass estimation in wheat (i) to calculate both DW and FW, (ii) based on the fusion of spectral and structural information at different processing levels in the designed novel workflow, (iii) to evaluate the developed approach against commonly used spectral VI-based approaches for modelling



biomass using simple regression technique, (iv) to show the model was developed on a wide range of genotypes with diversity in growth and development, and (v) to demonstrate the developed model relationship is valid across different growth stages including during and post-reproductive periods. Additionally, the study reports the generation of several intermediate traits including crop height model (CHM), crop coverage (CC) and crop volume (CV); which by themselves are capable of inferring important agronomic insights in high-throughput breeding research for screening genotypes.

## 2. Materials and Methods

In this study, three intermediate traits were mapped from multi-temporal UAV imagery, including CHM, CC and CV for modelling DW and FW biomass. While not previously explored for deriving DW and FW in breeding research, including in wheat using UAV data, these traits were selected for being physically related. A fusion-based image analysis approach was used to assess these traits at the individual plot level.

### 2.1. Field Experiment

The field experimental area was located at Agriculture Victoria's Plant Breeding Centre, Horsham, Victoria, Australia. This location has a mild temperate climate with an annual average rainfall of 448 mm and has predominantly self-mulching vertosol soil. The experiment comprised 20 wheat genotypes with four replications, each planted in 5 m × 1 m plot, with a density of approximately 150 plants per m$^2$. These genotypes were selected with diversity in growth patterns and biomass accumulation. The seeds for wheat genotypes were obtained from the Plant Phenomics' Grains Accession Storage Facility at Grains Innovation Park, Horsham. The list of wheat genotypes is provided in Supplementary Table S1.

### 2.2. Aerial Data Acquisition System and Field Data Collection

A custom data acquisition system was developed as one of the AgTech SmartSense products at SmartSense iHub, Agriculture Victoria. A MicaSense RedEdge-M multispectral camera (MicaSense, Seattle, WA, USA) was integrated with a DJI Matrice 100 quadcopter (DJI, Shenzhen, China) (Figure 1a). The Matrice's camera and gimbal operate on 12 V, so a switching mode power supply was used to step-down the output to 5 V for the MicaSense RedEdge-M sensor system. The system consists of a gravity-assisted 3D printed gimbal bracket and the original vibration dampeners. The system records the position values, i.e., latitude, longitude and altitude on to the camera tags, using the included 3DR uBlox global positioning system (GPS) module. Additionally, the multispectral camera also logs dynamic changes in incident irradiance levels using a downwelling light sensor (DLS). The sensor system was set to trigger and acquire images at the desired flying height of 30 m to achieve a ground sampling distance (GSD) of 2 cm. Image acquisition was set to automated capture mode with acquisition triggered via the GPS module at 85% forward and side overlap. The MicaSense RedEdge camera has five spectral bands: blue (475 nm), green (560 nm), red (668 nm), red edge (717 nm), and near-infrared (840 nm). A radiometric calibration panel with known radiometric coefficients for individual multispectral bands was used. Radiometric calibration measurements were recorded with the multispectral sensor before individual flight missions for image correction.

Five aerial imaging flight missions were undertaken at 30, 50, 90, 130 and 160 days after sowing (DAS) (Figure 1b). A range of in situ data was collected concurrently to support the UAV-based imagery, including ground control points (GCPs) for the geometric correction of the UAV and laying out radiometric calibration panels. Five permanent GCPs were installed and the position was recorded using a multi-band global navigation satellite system (GNSS) based real-time kinetic (RTK) positioning receiver (Reach RS2, Emlid Ltd., Hong Kong) with centimetre level precision. The estimated accuracy of the GNSS-RTK system was 0.02 m in planimetry and 0.03 m in altimetry. One GCP was placed in the centre and others at the four corners of the experiment area. Several ground truth measurements were undertaken. Plant height was manually measured by randomly selecting four wheat plants from each

experimental plot and averaging four height measurements to one per plot. The field experiment had 80 plots (four replicated plots of 20 wheat genotypes) and plant height was measured at the two time points on 130 DAS and 160 DAS, totalling 160 ground truth plant height observations. Plant height was measured from the ground level to the highest point of the plant, with the average of the four height measurements used as a representative ground truth measurement for the corresponding plot. Manual harvesting of plots was performed at four time points (50, 90, 130 and 160 DAS) for measuring biomass. The harvested samples were weighed upon harvest to measure FW and oven dried at 70 °C for 5 days to measure DW.

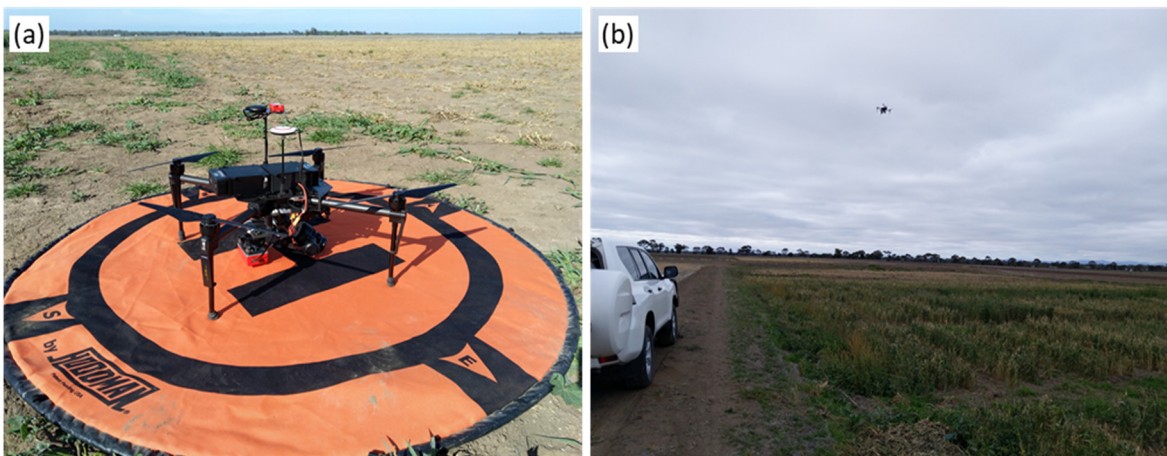

**Figure 1.** Field high-throughput phenotyping: (**a**) Unmanned aerial vehicle mounted with a RedEdge MicaSense multispectral camera, a global positioning system and downwelling light sensor; (**b**) Aerial mission on a field experiment at Plant Breeding Centre, Horsham, Victoria, Australia.

*2.3. Reflectance Orthomosaic, Digital Surface Model and Digital Terrain Model*

A processing pipeline was developed to process aerial multispectral images for modelling DW and FW (Figure 2). After the acquisition of raw images from the UAV-based multispectral sensor, the data were geometrically and radiometrically corrected using Pix4D Mapper (Pix4D SA, Lausanne, Switzerland).

Multispectral imaging sensors measure at-sensor radiance or the radiant flux received by the sensor. At-sensor radiance is a function of surface radiance (flux of radiation from the surface) and atmospheric disturbance between surface and sensor [53], often assumed to be negligible for UAV-based surveys [54]. However, surface radiance is highly influenced due to incident radiation. The multispectral sensor records at-sensor radiance measurements for each band as dynamically scaled digital numbers (DNs) at a determined bit depth. The conversion of the DNs into absolute surface reflectance values, a component of the surface radiance independent of the incident radiation (ambient illumination), is required if cross site, sensor, or time analysis is needed.

A common method for converting image DNs into absolute surface reflectance for UAV-based surveys includes an empirical line approach, whereby standard reflectance panels are used to establish a linear empirical relationship between DNs and surface reflectance under consistent illumination conditions during the survey [42,55–58]. Under varying illumination conditions, an extension of the empirical line approach is suited, whereby the logs from onboard DLS sensors are used to account for changes in irradiation during the flight [59–61]. The latter approach was used in this study to radiometrically correct the images with the inbuilt workflow in Pix4D Mapper, prior to further processing. Additionally, corrections were performed to rectify optical (filters and lenses) aberrations and vignetting effects to maintain a consistent spectral response [53,62].

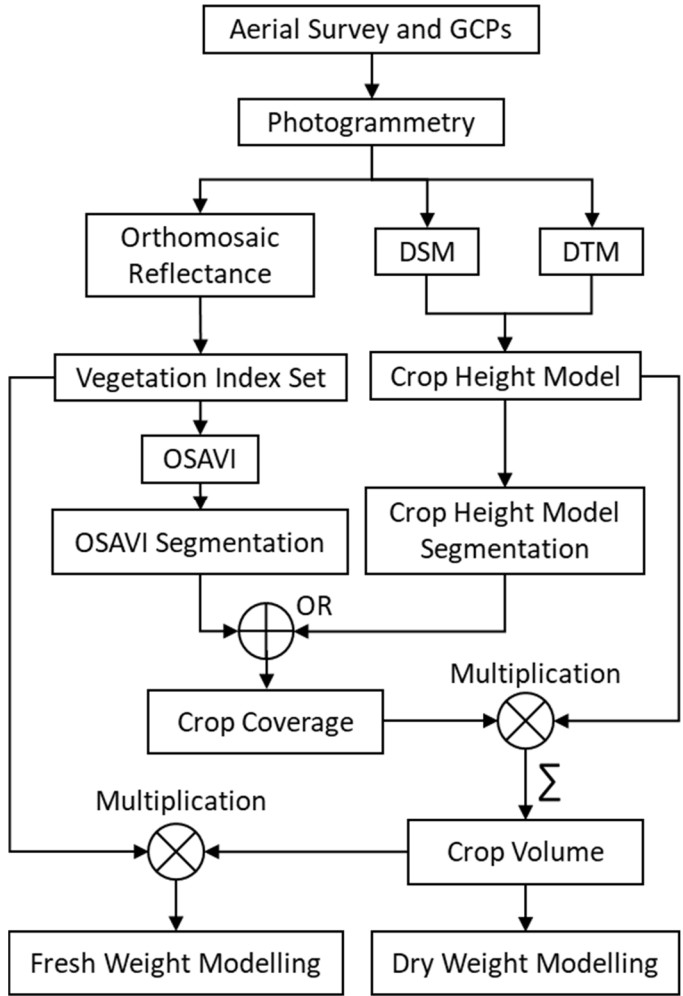

**Figure 2.** The processing workflow used for: Modelling—dry weight and fresh weight; Computing intermediate traits—crop height model, crop coverage and crop volume; Using—vegetation index, digital surface model (DSM) and digital terrain model (DTM).

Composite reflectance orthomosaic, digital surface models (DSM), and digital terrain model (DTM) images were generated by stitching together hundreds of calibrated images captured from individual flight missions in Pix4D Mapper. The software uses a popular structure-from-motion (SfM) technique [63], suited to combine a large number of images from UAV missions [64]. The process includes a scale-invariant feature transform (SIFT) [65] feature matching to create optimized resection geometry for improving initial camera position accuracy obtained through recorded GPS tags. Camera parameters are optimized based on matched triangulated target points between multiple images—in this case, the number of matched points set at 10,000 points was found to be sufficient for seamless optimisation as stitching. Bundle adjustment was then performed to generate a sparse model, containing generalized keypoints used to connect the images. The SfM algorithm subsequently adds fine-scale keypoints to the model during the reconstruction of a dense model, crucial in improving geometric composition. Known GCPs collected during the aerial survey were also used in the process to geometrically register the model. The entire SfM workflow inherently runs on a selected 'reference' band, with the selection of the correct 'reference' band being important in achieving better mosaicking results [7,66]. In this case, the 'green' bands of the multispectral images were used as a 'reference band' as the scene consisted primarily of vegetation features. Identical features in the overlapping portion of the images are connected using the computed keypoints to produce a composite reflectance orthomosaic, exported in rasterized (.tif) format. Bundle block adjustment was recomputed to optimize

the orientation and positioning of the underlying densified point cloud. The process yields a DSM and DTM of the study area, also exported in rasterized (.tif) format. The sigma of absolute geolocation variance error was 0.13 m, 0.33 m, 0.21 m, respectively, in x, y and z directions. The sigma of relative geolocation variance accuracy was 100%. Noise filtering was applied in the process of DSM and DTM generation, and a sharp surface smoothening was used to retain crop surface boundaries. All the exported layers, namely orthomosaic, DSM and DTM were resampled using an inverse distance weighting [67] function to 2 GSD, for consistency in pixel dimensions.

### 2.4. Image Processing and Data Analysis

The spectral and structural (DSM and DTM) layers obtained from the multispectral sensor were used to compute different intermediate layers which were fused at multiple processing levels (Figure 2).

### 2.4.1. Vegetation Indices (VIs)

The othomosaic spectral reflectance images were used to generate VIs, a spectral transformation of two or more multispectral bands to highlight a property of plants. This permits a reliable spatial and temporal inter-comparison of crop photosynthetic variations and canopy structural profiles. VIs are immune from operator bias or assumptions regarding land cover class, soil type, or climatic conditions, therefore suitable for high-throughput phenotyping. Additionally, seasonal, interannual, and long-term changes in crop structure, phenology, and biophysical parameters can be efficiently monitored. However, several multispectral VIs are prone to saturation issues which limit the extent to which they are suitable to infer the crop traits [32,68,69]. Furthermore, this requires a good understanding of the association of the multispectral bands with crop property for the selection of proper band combinations, index formulation and type of fitting function [70]. In this study, a set of 12 traditional VIs related to physiology and canopy structure was computed using the equations listed in Table 1.

**Table 1.** List of commonly used vegetation indices relating to physiology and canopy structure.

| Indices | Equation | Reference |
|---|---|---|
| Normalized difference vegetation index | $NDVI = \frac{NIR-RED}{NIR+RED}$ | [10] |
| Enhanced vegetation index | $EVI = \frac{2.5(NIR-RED)}{(NIR+6\times RED-7.5\times BLUE)+1}$ | [71] |
| Green normalized difference vegetation index | $GNDVI = \frac{NIR-GREEN}{NIR+GREEN}$ | [72] |
| Normalized difference red-edge index | $NDRE = \frac{NIR-RE}{NIR+RE}$ | [73] |
| Renormalized difference vegetation index | $RDVI = \frac{NIR-RED}{\sqrt{NIR+RED}}$ | [74] |
| Optimized soil adjusted vegetation index | $OSAVI = 1.6\left[\frac{NIR-RED}{NIR+RED+0.16}\right]$ | [12] |
| Modified simple ratio | $MSR = \frac{(NIR/RED)-1}{\sqrt{(NIR/RED)+1}}$ | [75] |
| Modified chlorophyll absorption ratio index 1 | $MCARI1 = 1.2[2.5(NIR - GREEN) - 1.3(RED - GREEN)]$ | [76] |
| Modified chlorophyll absorption ratio index 2 | $MCARI2 = \frac{3.75(NIR-RED)-1.95(NIR-GREEN)}{(2\times NIR+1)^2-\left(6\times NIR-5\ \sqrt{RED}\right)-0.5}$ | [76] |
| Modified triangular vegetation index 1 | $MTVI1 = 1.2[1.2(NIR - GREEN) - 2.5(RED - GREEN)]$ | [76] |
| Modified triangular vegetation index 2 | $MTVI2 = \frac{1.8(NIR-GREEN)-3.75(RED-GREEN)}{\sqrt{(2\times NIR+1)^2-\left(6\times NIR-5\ \sqrt{RED}\right)-0.5}}$ | [76] |
| Pigment specific simple ratio for chlorophyll a | $PSSRA = \frac{NIR}{RED}$ | [68] |

The selection of appropriate VIs is critical in high-throughput plant phenotyping studies aiming for high performance. Studies need to integrally examine the suitability of the VIs for the intended application and objective. Inappropriately selected VIs can produce inaccurate results burdened with a number of uncertainties [77]. For estimating biomass, VIs in this study were selected from existing literature, such as NDVI, enhanced vegetation index (EVI), green normalized difference vegetation index (GNDVI), normalized difference red-edge index (NDRE), renormalized difference vegetation index (RDVI), optimized soil adjusted vegetation index (OSAVI) and modified simple ratio (MSR). A few VIs were also selected which correspond to biophysical and biochemical parameters, such as modified chlorophyll absorption ratio index 1 (MCARI1), modified chlorophyll absorption ratio index 2 (MCARI2), modified triangular vegetation index 1 (MTVI1), modified triangular vegetation index 2 (MTVI2), and pigment specific simple ratio for chlorophyll a (PSSRA).

### 2.4.2. Crop Height Model (CHM)

A pixel-wise subtraction of the DTM altitudes from DSM altitudes was performed to generate a CHM, representing the relief of the entire crop surface. The pixel-wise subtraction step essentially means that the computation is performed between corresponding pixels of source DTM and DSM layers. The accuracy of CHM computed using SfM approach, relies on interacting factors including the complexity of the visible surface, resolution and radiometric depth, sun-object-sensor geometry and type of sensor [78]. The canopy surface for wheat is very complex containing reflectance anisotropy and micro-relief height variation. A $3 \times 3$ pixel local maximum moving filter was applied on the CHM layer to enhance the highest peaks and reduce the micro-variation. The implemented filter moves the pre-defined window over the CHM and replaces the centre pixels value with the maximum value in the window, if the centre pixel is not the maximum [79,80].

### 2.4.3. Crop Coverage (CC)

Previous studies have used different VIs to classify CC or vegetation fraction, the vegetation part of the research plots [81–83]. In this study, the optimized soil adjusted vegetation index (OSAVI) was used for its ability to suppress background soil spectrum to improve the detection of vegetation. Both OSAVI and CHM were used to create a CC layer to mask the extent of the vegetation for individual plots across all time points. Firstly, individual segmentation layers were prepared for OSAVI and CHM using a dynamically computed threshold using the Otsu method, an adaptive thresholding approach for binarization in image processing [84]. This threshold is computed adaptively by minimizing intra-class intensity (i.e., index values for OSAVI and height levels for CHM) variance, or equivalently, by maximizing inter-class variance. In the simplest form, the algorithm returns a single threshold that separates pixels into two classes, vegetation and background. The technique was suitable to filter unwanted low-value OSAVI and CHM pixels, corresponding to minor unwanted plants such as weeds or undulated ground profile, respectively. Finally, the pixel-wise product of segmented OASVI and CHM pixels was used to prepare the CC mask corresponding to vegetation, in this case, wheat. The rationale for the adopted approach is that (i) OSAVI utilized the 'greenness' of wheat to detect its mask, 'greenness' drops during flowering and after maturity, and (ii) CHM utilizes the crop relief to detect the corresponding mask, thereby it is immune to changes in 'greenness' so applicable during flowering and post-emergence of maturity, but suffers when the plants are too small (less than 5 cm approximately) as the crop canopy is very fragile at this stage for the photogrammetric SfM approach to generate a dependable CHM. These independent issues were resolved through the fusion of OSAVI segmentation and CHM, improving classification of the CC of wheat in research plots.

A rigorous approach to evaluate the achieved classification accuracy for the CC layer was performed through a comparison of CC classified labels against ground truth across randomly selected locations using a confusion or validation matrix. Over the five time points, a total of 1500 ground truth points (i.e., 300 points in each time point) were generated between the two classes—wheat CC and ground—using an equalized stratified random method, creating points that were randomly distributed

within each class, where each class has the same number of points. The ground truth corresponding to each point for validation was captured manually through expert geospatial image interpretation training using high-resolution (2 cm) RGB composite orthomosaic images. Accuracy measures namely producer's accuracy, user's accuracy, overall accuracy (OA) and kappa coefficient (K) were computed using the confusion matrix. In traditional accuracy classification, the producer's accuracy or 'error of omission' refers to the conditional probability that a reference ground sample point is correctly mapped; whereas the user's accuracy or 'error of commission' refers to the conditional probability that a pixel labelled as a class in the map belongs to that class [85]. Overall accuracy refers to the percentage of a classified map that is correctly allocated, therefore is used as a combined accuracy parameter from a user's and producer's perspective [85]. A kappa value ranges between 0 and 1, representing the gradient of agreement between 'no agreement' and 'perfect agreement'.

2.4.4. Crop Volume (CV) and Dry Weight (DW) Modelling

An intermediate metric, CV was computed by multiplying CHM and CC, followed by summing the volume under the crop surface. The multiplication of CHM with the CC layer aids in mitigating against ground surface undulations and edge-effects in surfaces reconstructed through SfM. Equation (1) depicts the mathematical formulation of the CV metric:

$$CV = \sum_{i=1}^{i=m} \sum_{j=n}^{j=n} CHM_{i,j} \times CC_{i,j} \tag{1}$$

where $i$ and $j$ represent the row and column number of the image pixels for an $m \times n$ image, size of an individual plot.

Classically, the mass (or generally called weight in applied sciences) (W) of a physical substance is directly related to its density (D) and volume (V) according to Equation (2). The case of physical modelling DW using CV is not as straightforward, as the measured volume parameter, CV is not exactly the 'volume of dry matter tissue' ($V_{tissue}$); instead, CV is compounded with the fraction of canopy fragility, i.e., the 'volume of air' ($V_{air}$) of empty space within a crop canopy grown in plots. Accordingly, Equation (3) can be modified to account for both $V_{tissue}$ and $V_{air}$. The equation could be further expanded to identify the proportional dependence of DW on CV, as the other 'unmeasured' factors could be ignored – $D_{tissue}$ being a constant for a selected crop type (i.e., wheat) and $V_{air}$ being a linearly scaled variable with CV for a given crop type in plots with constant sowing rate. The proportionality could be refactored as a simple linear regression model Equation (4), whereby the coefficients, slope ($\alpha$) and bias ($\beta$), could be calculated parametrically using measured (ground truth) DW values. This expression provides the modelled relationship to derive DW using CV, through non-invasive and non-destructive means applicable for field high-throughput phenotyping.

$$W = D \times V \tag{2}$$

$$\begin{aligned} CV &= V_{tissue} + V_{air} = \frac{DW}{D_{tissue}} + V_{air} \\ &=> DW = D_{tissue}(CV - V_{air}) \\ &=> DW = C(CV - V_{air}), \text{ as } D_{tissue} \text{ could be considered constant (C) for wheat} \\ &=> DW \propto CV \text{ , by ignoring the constant C and } V_{air}, \text{ being a linearly scaled variable with CV} \end{aligned} \tag{3}$$

$$DW = \alpha \cdot CV + \beta \tag{4}$$

2.4.5. Crop Volume Multiplication with Vegetation Indices (CV×VIs) and Fresh Weight (FW) Modelling

To model FW, the intermediate metric, CV was multiplied with the set of derived VIs. The rationale for this approach is that, (i) CV is a canopy structural metric computed through SfM approach as such is able to estimate the dry tissue content or DW, but is void of the ability to infer the fresh tissue water

content or FW, (ii) VIs on the other hand, are reflectance derived biophysical parameters having the ability to infer photosynthetic variations and related water content in vegetation. The mathematical product of CV and VIs is deemed to resolve the limitations of individual parameters (Equation (5), modified after Equation (4)). It is imperative to state that the linear regression model coefficient values will vary corresponding to the different VIs.

$$FW = \alpha \cdot CV \times VIs + \beta \tag{5}$$

### 2.4.6. Plot Level Data Analytics

A shapefile (.shp) consisting of the individual field plot information was created in ArcMap version 10.4.1 (Esri, Redlands, CA, United States). The image processing and analytics involved beyond Pix4D processing and shapefile creation in ArcMap, i.e., generation of VIs, CHM, CC, CV, CV×VIs, DW and FW, were all computed in Python 3.7.8 (Python Software Foundation. Python Language Reference) using source packages including os, fnmatch, matplotlib, numpy, fiona, shapely, skimage, opencv2, rasterio, and geopandas. The coded workflow involved the generation of the intermediate geospatial layer corresponding to individual traits, clipping the layers to plot geometries, summarizing the traits in individual plots, analyzing and validating the summarized traits.

## 3. Results

### 3.1. Crop Height Model

Plant height of wheat genotypes in the experiment ranged from 54 to 91 cm on 130 DAS and 62 to 98 cm on 160 DAS with a normal distribution. The mean plant heights were 71.8 and 78.9 cm on 130 DAS and 160 DAS, respectively. To evaluate SfM-derived CHM's performance with respect to ground truth plot height measurements, a correlation-based assessment was performed (Figure 3). The assessment achieved a strong and statistically significant (*p*-value < 0.001) linear relationship between CHM and ground truth plant height with a coefficient of determination ($R^2$) of 0.81 and a standard error estimate (SEE) of 4.19 cm. To minimize differences due to plant growth, the field measurements were carried out on the same days of the aerial surveys. Unlike the highest points measured during ground-based surveys, the CHM represents the entire relief of the crop surface; therefore, the average CHM was found to be about 23.5 cm lower than the actual average canopy height.

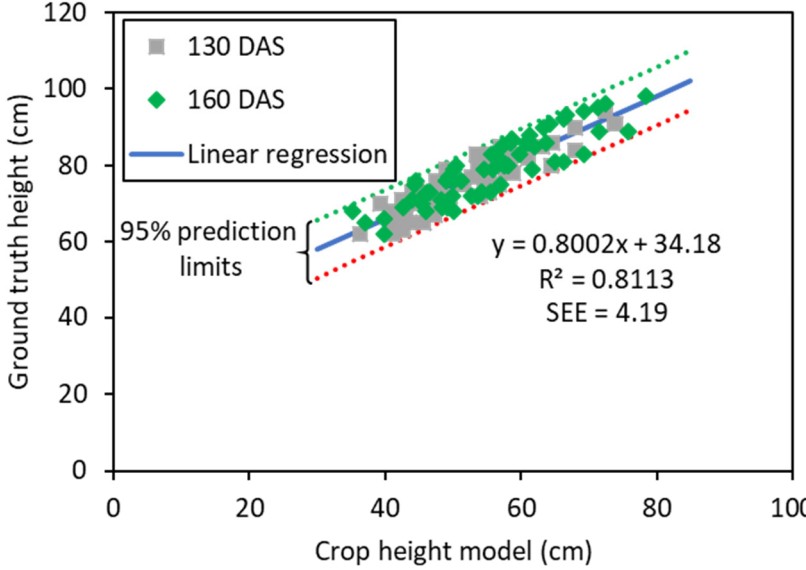

**Figure 3.** Correlation between computed crop height model and observed ground truth height measurements at two time points.

### 3.2. Crop Coverage

To validate the fusion-based algorithm, Figure 4 shows the classified CC and CC × VIs images against RGB composite images generated using multispectral bands blue (475 nm), green (560 nm) and red (668 nm). A section of the entire field trial was focused on to include three genotypes with different fractional density, plant height and variable growth stage effects. The proposed algorithm performed well throughout the vegetation portion of the plot with minor classification challenges around the edges of vegetation. The characteristics of vegetation and ground are quite similar along the borders, so it was difficult to achieve good results in these areas. Additionally, the fractional nature of wheat canopy also affected the accuracy, due to the presence of noisy pixels comprising of mixed spectra from both wheat and ground; and the limitation of the SfM approach to resolve the finer details along high-gradient relief variations.

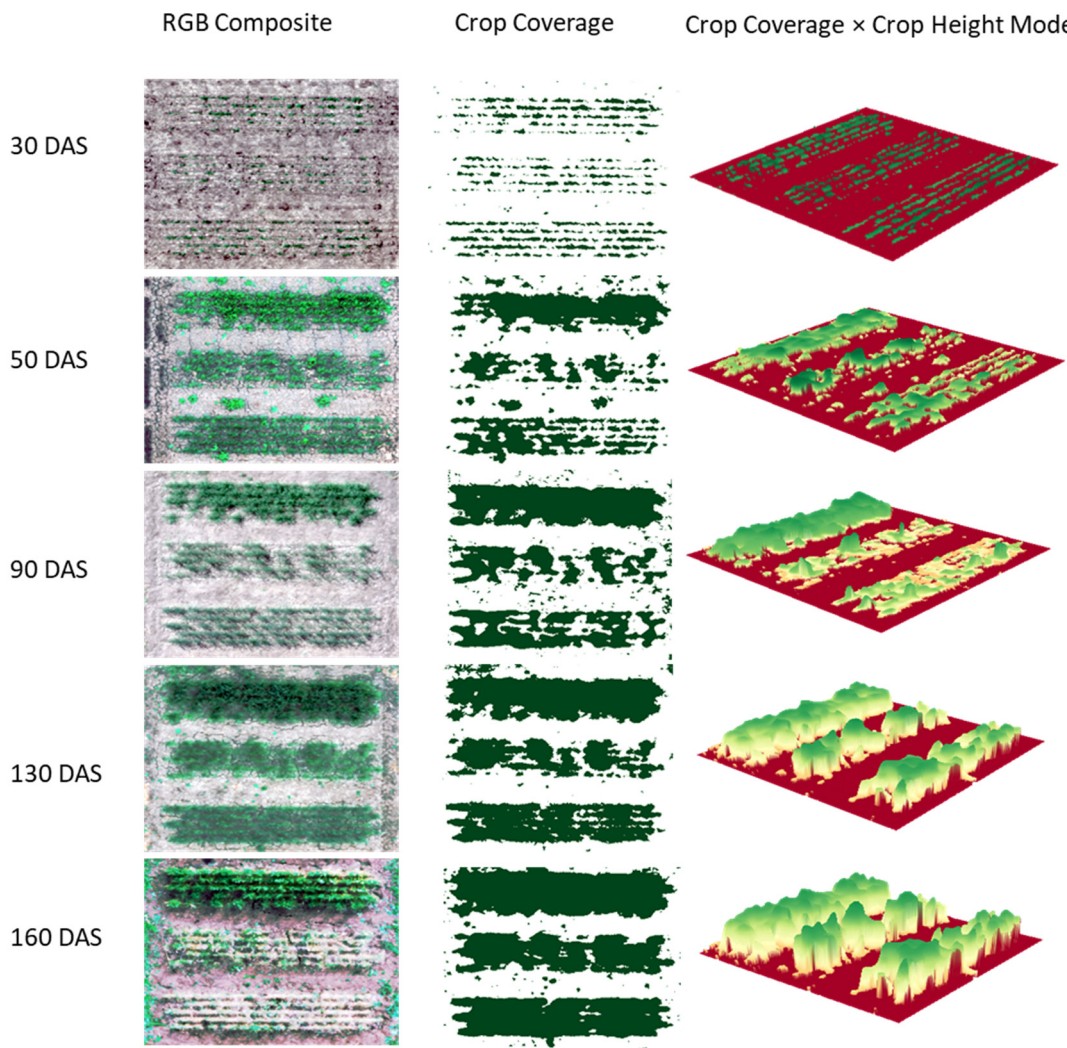

**Figure 4.** A visual evaluation of the performance of the employed workflow for computing intermediate metrics—crop coverage (CC) and crop volume (CV) at different growth stages (30, 50, 90, 130 and 160 DAS) on three genotypes. The classified CC and CC × VIs images are compared against RGB composite images generated using multispectral bands blue (475 nm), green (560 nm) and red (668 nm).

The classification class CC achieved a user's accuracy of 98.9%, a producer's accuracy of 99.4% and an OA of 99.2%. In addition to the traditional estimates, the classification achieved a K of 0.98, which is an indicator of agreement between classified output and ground truth values.

### 3.3. Dry Weight Modelling

The proposed approach, of modelling DW using CV, was compared against a traditional VI-based approach. The linear regression demonstrated that the degree of correlation (in terms of $R^2$) for the VI-based approach in modelling DW become less accurate at progressive time points (>50 DAS); while the accuracy of CV-based approach for modelling DW remained consistently high (Figure 5a). For the interpretation of the results, correlations were considered hereafter as low ($R^2 < 0.70$), moderate ($0.70 \leq R^2 < 0.85$) and high ($R^2 \geq 0.85$). These results were influenced by the fact that experimental plots of diverse wheat genotypes typically have variable growth dynamics, common in crop phenotyping research; therefore, exhibiting variation in reflectance responses which is limiting to the accuracy of the linear model. The CV, on the other hand, is derived through structural means using SfM and is immune to variation in the spectral properties of plants over different growth stages across research plots. Of the four time points studied in our experiment for biomass estimation, the earliest time point showed genotypes in different plots were in overall similar growth stages, as such the VI derived model prediction was highly correlated. At later time points, genotypes attained different growth stages influencing the spectral signatures and the VIs-derived correlation was low. When data from all the time points in the experiments were combined, CV was still accurately predicting DW, whereas traditional VIs exhibited low $R^2$ values (Figure 5b). This observation was interesting as it seems contradictory to previous studies which demonstrated the high prediction potential of VIs, particularly related to structural characteristics of plants, i.e., NDVI, GNDVI, OSAVI, MTVI1 and MTVI2 [11,86–88]. However, these previous studies had established a successful relationship with biomass under restricted complexities, i.e., either on a single genotype in a field or in a single time point. The experimental work conducted in this study indicates that CV derived using UAV-based SfM information is consistent with genotypic variation and across different time points, as an accurate predictor ($R^2 = 0.96$ and SEE = 69.2 g/m$^2$) of DW with statistical significance (*p*-value <0.001) (Figure 5c).

### 3.4. Fresh Weight Modelling

The approach of modelling FW using CV×VIs was evaluated using different CV and VI combinations, and against independent VIs. For evaluating different CV×VIs combinations, a plot of the corresponding $R^2$ values obtained for predicted vs. observed FW values across different time points was used (Figure 6a). The performance of three combinations, CV×GNDVI, CV×EVI and CV×NDRE were found to be consistently higher across different time points, compared to the other CV×VI combinations. The performance of CV×EVI was strongest amongst the three CV×VIs combinations for FW estimation. When data from all four time points across the experiment for FW were combined, the CV×EVI was again found to outperform other CV×VIs (Figure 6b). Therefore, CV×EVI was used as an estimator to model FW computation (Figure 6c). The estimator achieved a strong ($R^2 = 0.89$ and SEE = 333.54 g/m$^2$) and statistically significant (p-value < 0.001) linear relationship with ground truth measurements of FW. Additionally, the regression analysis demonstrates that the $R^2$ for the original VI-based approach in modelling FW increases significantly for all VIs when coupled with CV (Figure 6b). The proposed fusion of CV×VIs takes advantage of both spectral and structural information provided by VIs and CV, respectively, whereas VIs were unable to account for structural variations between different genotypes.

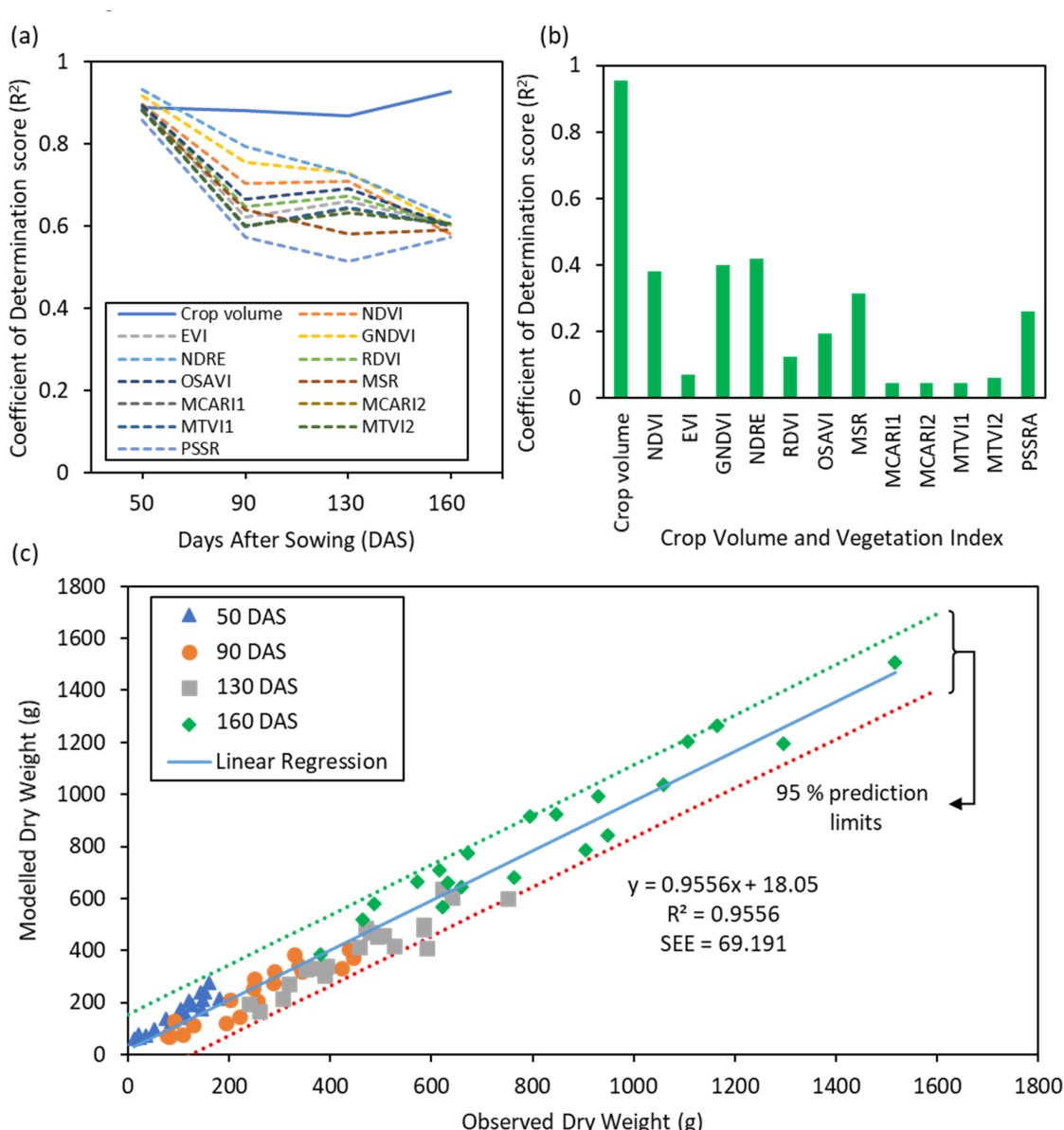

**Figure 5.** Estimation of dry biomass or dry weight (DW): Variability in the correlation of determination ($R^2$) for modelling performed using crop volume (CV) and standard vegetation indices (**a**) across different time points and (**b**) at all the time points combined against observed DW; (**c**) Modelling results for DW derived from CV against observed DW data.

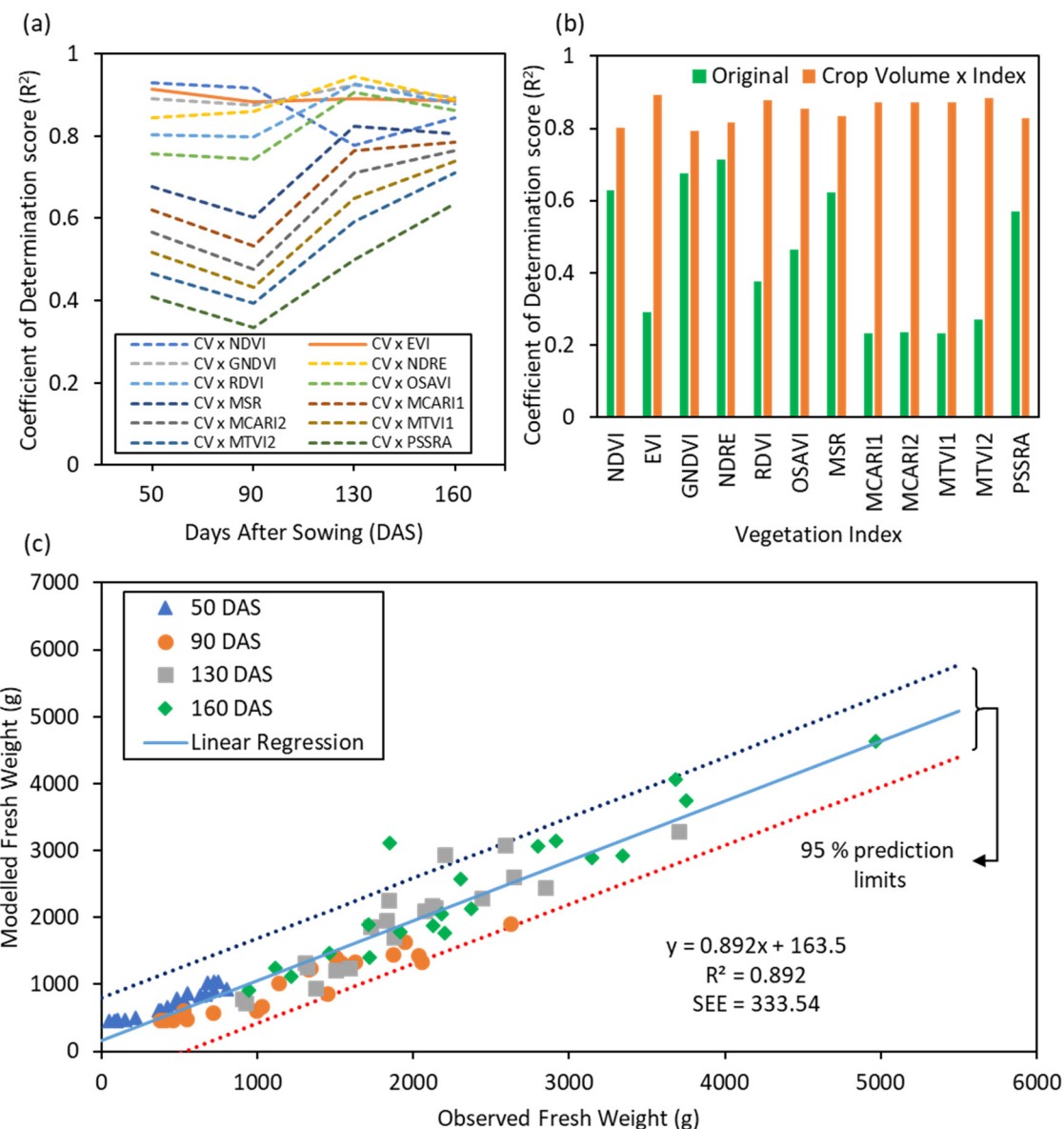

**Figure 6.** Estimation of fresh biomass or fresh weight (FW): Variability in the correlation of determination ($R^2$) of crop volume multiplied using standard vegetation indices (CV × VIs) (**a**) across different time points and (**b**) at all the time points combined against observed FW; (**c**) Modelling results for FW derived from CV × EVI against observed FW readings.

### 3.5. Genotypic Ranking Using Ground Measurements and UAV-Based Measurements

The objective of this study was primarily to develop an accurate, non-invasive, UAV-based, analytical framework to derive DW and FW for wheat genotypes. DW and FW estimated across the four time points showed expected and consistent growth trends for wheat genotypes (Figure 7). Most genotypes showed a steady growth of DW until the third time point, followed by nearly exponential growth between the third and fourth time point (Figure 7a). The FW, on the other hand, followed a steady growth until the fourth time point (Figure 7b). Importantly, the derived multitemporal biomass models were able to capture these trends in plant development. The two biomass models were able to illustrate different growth dynamics seen between diverse wheat genotypes. Genotypes Sunvale, Volcani DDI, Gladius, Ellison, Hartog, Ventura, Carnamah, EGA Gregory, Kennedy and Sunco demonstrated an early vigour, producing comparatively more biomass during early to mid-vegetative

growth. Overall, Carnamah produced the most biomass, while Derrimut produced the lowest DW and FW.

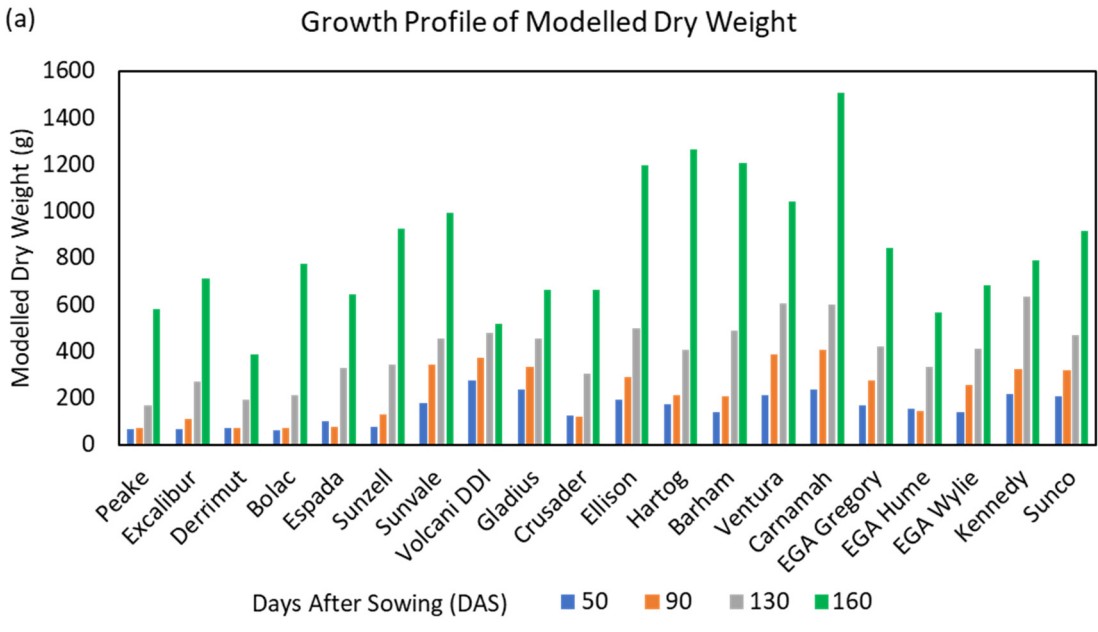

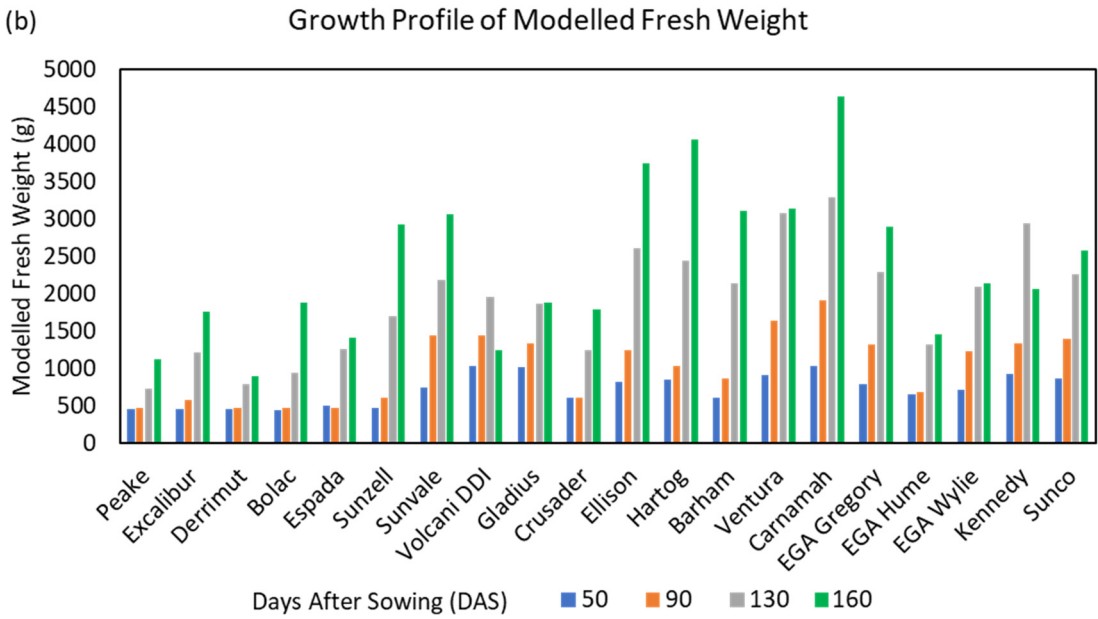

**Figure 7.** Growth profile of wheat genotypes at different time points. Using modelled (**a**) Dry weight and (**b**) Fresh Weight.

## 4. Discussion

Modern plant breeding strategies rely on high-throughput genotyping and phenotyping. Conventional phenotyping relies on manual and/or destructive analysis of plant growth measures such as biomass [89]. To investigate the growth rate of a diverse number of genotypes in field breeding research, a large number of plants needs to be harvested regularly over different growth stages; a method which is very time consuming and labour intensive. Moreover, as this method is destructive, it is impossible to take multiple measurements on an entire plot at different time points. To some

extent, this is addressed by harvesting a subset number of plants or a section of field plot each time, but might not be representative of the entire plot or suffer from selection bias. Accuracy improvements can be made by adding replicated plots for individual genotypes for each time point harvest, although this might not be economically or spatially feasible over large populations; whereas high-throughput imaging technologies are non-destructive, allowing for repetitive measurements of the same plants, and are cost effective and efficient compared to traditional phenotyping, making them important for crop breeding programs [90].

Biomass measures of DW and FW are challenging under field conditions, especially when studying a wide range of genotypes with diverse growth patterns, including during reproductive and senescence stages [15,45,91]. In this study, the fusion of complementary spectral and structural information derived using the same multispectral sensor was found to be a suitable and robust alternative to traditional methods involving purely spectral VIs. The fusion-based workflow developed here describes research utilizing the fundamental information generated through a UAV-multispectral sensor to generate several intermediate metrics (i.e., VIs, CHM, CC, CV). The interaction between these intermediate metrics was used at different levels to improve the accuracy at successive stages, thereby helping in developing a model relationship for DW and FW. For example, the fusion was achieved through (i) the logical 'OR' operation between OSAVI segment and CHM segment layer to derive CC, (ii) the mathematical product between CC and CHM followed by summation over the plot area to calculate the CV relating to DW, and (iii) the multiplication between CV and VIs to retrieve FW (Figure 2).

The intermediate metrics resulting from the processes of deriving biomass, namely CHM and CC, are themselves beneficial. Traditionally, for short crops such as wheat, plant height is measured using a ruler in the field by selecting a few representative plants to represent the canopy height. The method is labour intensive, time-consuming and expensive for large breeding trials. Measuring variation in crop height associated with growth at finer temporal rates remains largely impractical in widely distributed field trials. High-throughput field imaging platforms have demonstrated usefulness in extracting crop height in each plot [43,44,92]. The CHM layer derived herein achieved satisfactory model correlation in estimating crop height in plots (Figure 3) and was found to be acceptable to measure the fine-level of crop height in a genotypically variable population. However, the SfM's matching accuracy for a crop surface is likely to be limited due to little-to-no or analogous texture (pattern created by adjacent leaves), surface discontinuities (gaps in canopies), repetitive objects (regularized plots), moving objects (such as leaves and shadows), multi-layered or occlusions (overlapping leaves) and radiometric artifacts [93–95]. Nevertheless, the achieved high accuracy in estimation of height in wheat plots was characterized by the low-oblique vantage of the images captured from UAVs, aiding in the accurate reconstruction of depth models.

The CC or crop fractional cover is an important phenological trait of crops, which can be used as an estimate of crop emergence in early growth stages, and an indicator of early vigour and crop growth rates during the vegetative growth stages [96]. The computed CC layer has been found to be more reliable and advantageous than subjective visual scores (Figure 4). The accuracy of the CHM and CC depends on the precise extraction of the DTM layer or the 'bare earth model'. The process of extracting DTM using the SfM algorithm involves measuring the lowest terrain altitude points in the scene, as seen by the sensor when the canopy is fragile or when there is a visible bare earth surface. Herein, sufficient inter-plot spacing has been found to benefit extraction of DTM, which thereby is advantageous for improving the accuracy of CHM and CC. Related factors such as the presence of high surface soil moisture after heavy rain or irrigation events, and presence of vegetation in inter-plot spacing, i.e., crop overgrowth or weeds, have been found to lower the accuracy of CHM. This is because the presence of both high moisture and inter-plot weeds limit the bare-earth reflectance signature, which in turn reduces the accuracy of DTM. Therefore, avoiding immediate UAV-flights for data acquisition after a heavy rain or irrigation, and ensuring a clean field trial is maintained, has been found to produce a reliable result.

In this study, a simple linear regression model relationship was used to demonstrate the efficacy of the presented CV and CV×VIs approach against other standard VIs based approaches. Multivariate analysis, or conventional regression techniques (multiple linear regression, multiple stepwise regression techniques and partial least squares regression) and machine learning (artificial neural networks, random forest regression and support vector machine regression) are potential methods which can combine multiple variables in model predictions. This aspect was beyond the scope of the presented work; nevertheless, these approaches can be adopted as the future potential to extend this study.

The central idea for the presented work is unique in the aspect that DW could be estimated using CV and FW using CV×VIs. The concept behind this approach could be understood by considering that with a constant plant density for a crop, the density factor ($D_{tissue}$) and air space within the canopy ($V_{air}$) could be assumed to be constant. Therefore, DW of the tissue should linearly correlate with CV ($R^2 = 0.96$ in Figure 5b,c). In the process, water is present inside the 'plant tissue', which does not significantly influence CV, which as such could not be used robustly to measure FW ($R^2 = 0.62$ in Figure 6b). VIs, on the other hand, are influenced primarily by the chlorophyll content or greenness of the plant, which in turn are influenced by photosynthetic potential and water uptake during the growth. In other words, high FW can imply high water uptake in tissue, better photosynthetic ability, and generally plants with more greenness corresponding to higher values of VIs. However, the amount of plant matter or CV undergoing the photosynthesis process remains unaccounted for from this notion. Therefore, multiplying CV with VIs fills the gap, increasing the simple linear model relationship in measuring FW ($R^2 > 0.8$ for all CV×VIs combinations in Figure 6b; with best $R^2 = 0.89$ for CV×EVI in Figure 5b,c).

For high-throughput phenotyping application in crop breeding research, it is important to have a metric which could be used as a proxy to a certain agronomic trait. In this study, CV is a proxy to DW, and CV×EVI a proxy to FW. Other studies similarly used VIs [21,26] or combination of structural and spectral information [43,51,52] in modelling biomass, which is also a proxy-based approach. It could not be claimed with the available data in the study that the reported simple linear model relationships would hold exactly same every year, nevertheless the metrics CV and CV×EVI could certainly be used as a proxy to screen crop genotypes with higher or lower biomass and monitor their growth patterns over time.

Previous studies employing proximal remote sensing technologies have largely used spectrally derived VIs to develop a model relationship with biomass [18,24,28,31]. During the reproductive growth stage, wheat heads or spikes lower the fraction of vegetative green cover of the plot, thereby influencing the spectral VI's ability to estimate biomass reliably. Therefore, yield models relying on estimated biomass using spectral VIs, are less or not effective during reproductive growth stages. Additionally, different genotypes of wheat have variable growth patterns influenced by genetic and environmental effects. Diverse genotypes exhibit different spectral profiles, correspondingly the VIs do not correlate with plot biomass during multi-growth periods (Figure 5a,b and Figure 6a,b). Previous studies have reported a similar observation, that methods based on visible and near-infrared bands underestimate biomass in multi-growth periods [43,97]. In particular, a model to monitor biomass and grain yield, that is valid only for a single growth stage, is of less importance in high-throughput phenotyping [31,37,39,98]. Therefore, the development of new methods was needed to improve the accuracy to estimate biomass across multiple growth stages [45], justifying the practical applications of the presented work. Previous studies have suggested combining spectral VIs and crop height [44,97], and have reported combining VIs and surface model [43] obtained from UAV images to improve the accuracy of biomass estimates. Other studies have demonstrated that combining spectral VIs with structural information derived from SAR and LiDAR to improve biomass and leaf area index estimates [38,99]. Comparing information from SAR and LiDAR to complement VIs, UAV-based structural (surface models) information are cost-effective for ranking of genotypes in field breeding experiments.

## 5. Conclusions

In this study, we examined the suitability of fusing spectral and structural information from UAV images for the estimation of DW and FW in a wheat experiment. Intermediate metrics including CHM, CC, CV were computed in the process of modelling DW and FW. The analysis showed that all the intermediate metrics and modelled DW and FW values were accurate and highly correlated with equivalent ground truth measurements. The results demonstrated that the proposed novel approach was robust across different growth stages, maintaining a single model relationship. Additionally, the approach outperformed widely used traditional methods of using multispectral VIs to estimate biomass. Furthermore, the mentioned approach is time-efficient and cost-effective compared to other methods employing secondary sensor systems such as SAR and LiDAR, in addition to multispectral sensors, or a dedicated hyperspectral sensor to compute narrow band VIs. The approach is also effective for ranking genotypes in breeding trials based on their biomass accumulation. This study suggests that the proposed approach of using imaging tools and analytics provides an important and reliable method of high-throughput phenotyping for faster breeding of better crop varieties.

**Supplementary Materials:** The following are available online at http://www.mdpi.com/2072-4292/12/19/3164/s1, Table S1: List of wheat genotypes used in the field trial.

**Author Contributions:** Conceptualization, B.P.B. and S.K.; methodology, software, validation, formal analysis, data curation, writing—original draft preparation, B.P.B.; investigation, resources, writing—review and editing, supervision, S.K.; project administration, funding acquisition, S.K. and G.S. All authors have read and agreed to the published version of the manuscript.

**Funding:** This research received no external funding.

**Acknowledgments:** We thank Nathan Good for conducting some UAV flights and pre-processing the multispectral data. Emily Thoday-Kennedy and Sandra Maybery for help in managing the field trial and manual measurement of crop height and biomass harvest.

**Conflicts of Interest:** The authors declare no conflict of interest.

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
