# Peer review of "Fusion of Spectral and Structural Information from Aerial Images for Improved Biomass Estimation"

_remotesensing, doi:10.3390/rs12193164_

Round 1
Reviewer 1 Report
This paper assesses the use of multi-spectral info from UAVs to accurately estimate crop biomass in wheat fields. The authors use a multi-spectral camera (MicaSense RedEdge-M) to derive orthomosaics and intermediate metrics (i.e. crop height model, crop coverage and crop volume) from canopy surface information (DSM-DTM), and subsequently fuse the two information layers to model biomass. Because the methodology combines both types of information (spectral and structural), the method seems quite accurate during reproductive and senescence stages.
The topic is current and fits the scope of this journal. The manuscript has an organized structure of research paper and it is written in a clear English. Overall, the introduction reads well, and it provides an adequate background for the research. The methodology is well designed but some details can be explained more clearly. The discussion is consistent with the results obtained. Summarizing, in my opinion, the paper is correct and interesting to read, thus it could be approved for publication in Remote Sensing without any substantial amendment. I would simply suggest the following minor reviews (presented in no particular order):
- Line 80. “In our knowledge, no study is available yet that evaluates the estimation of biomass using a fusion of spectral and structural from multispectral images captured from small UAVs”. I am not completely sure about this (e.g. org/10.3390/rs12142199, doi.org/10.1016/j.jag.2015.02.012, etc.). I would suggest rewriting the sentence in a more precise way.
- The sentences (Line 138-139) “Plant height was measured […] four representative wheat plants” and (Line 141-143) “[…] with the average of the four height measurements […]” sounds repetitive. Please, specify how representative plants have been selected. Selection bias?
- Line 175. “The software uses a popular structure-from-motion (SfM) technique”. Requires citation (e.g. https://doi.org/10.1016/j.geomorph.2012.08.021). Although the field of application is different, this reference is probably the most appropriate since SfM is defined for the first time in this paper. There are other works that apply SfM to biomass estimation more related to this study that can also be discussed (e.g. doi.org/10.3390/rs11222678, doi.org/10.1080/10106049.2018.1552322, etc.).
- Line 180: “the number of matched points was set at 10,000 points”. Requires further justification. Please, also include more info about the images of the dataset used (size in pixels, GSD, flight height, etc.).
- Line 191. What is the horizontal/vertical accuracy of the DSM and DTM?
- Line 220. “Pixel-wise subtraction”. Please, explain in more detail how this procedure is performed. Is it an automatic procedure or are the points selected manually from "bare field"?
- Line 451. “was characterized by the low-oblique vantage of the images captured from UAVs”. Please, clarify what you mean by low-oblique and explains why it is advantageous.
- - Some details should be corrected (e.g. please, include year in [30] and [68], year is duplicated in [16], [17], [35], [36], [39], etc.).
Author Response
Uploaded response in word file

Reviewer 2 Report
Dear authors,
I found your article quite attractive, the idea of a fusion of spectral and structural information is always somehow interesting and potentially beneficial. The idea and the structure of your paper are clear, I have a few comment and suggestions only.
I wouldn’t introduce CSM abb. (in the Introduction) and use CHM instead as you use it later at the paper. In the Methods you introduce DEM abb. while later in the paper you use DTM instead, please unify those terms. You talk about the photogrammetry while you describing SfM, I would be a little bit circumspect about that, I would prefer some other term instead – photo-reconstruction, image-matching etc.
How did you choose the VIs? Many of them tend to be correlated, did you inspect VIs intercorrelation? Many studies exist on that topic (e.g. https://doi.org/10.7717/peerj.5487).
You introduce 2 cm DSM as well as DTM using Pix4D, however, the resolution of DTM used to be much lower than DSM. Why you resampled DTM later?
In my opinion, some of part are described too detailed, e.g. chaps. 2.3, 2.4.2, 3.2 – describing of errors; and some parts, which are quite easy and clear, are described a little bit complicated, e.g. 2.4.3.
You found 34 cm of the systematical shift of your CHM. Couldn't this be introduced partly by sharp filtering you conduct?
Introducing those accuracies, Table 2 is useless, in my opinion.
The discussion is sometimes a little bit hard to follow. I would suggest separating your three parts into more consistent paragraphs.
You describing the uncertainties of DTM generation via reflectance signatures (moisture, shadows etc.). That's generally true, but it is not the case of Pix4D.
All the best.
Author Response
Uploaded response in word file

Reviewer 3 Report
The submitted manuscript presents a very simple yet apparently quite effective approach to estimating crop dry and fresh weight from UAV survey data (spectral and
derived DSM). This is interesting and useful as it demonstrates the utility of UAVs in a specific (and important) application context. The results are not
groundbreaking or very surprising but worth reporting, and Remote Sensing is a suitable publication outlet for this.
My greatest concerns are related to the methodology. It is really quite simple and should be easy to understand, but it is poorly presented in the Methods section.
Additional critical analyses and considerations should also be included in the Results and discussion, e.g. is the model performance genotype-dependent, and how
transferable to different growing seasons and flight campaigns (with different biases e.g. in crop height estimation) is this model?
My recommendation is to accept this manuscript subject to major modifications, and to carefully re-evaluate a revised manuscript.
Detailed comments:
Title and throughout the paper: "Fusion" suggests that data fusion is done, e.g. in multiple-variable empirical models that combine predictors from different data
sources. "Combination" seems more appropriate here.
Abstract: The abstract lacks important information, e.g. a bit more information on the kind of model that was used to model CV, CHM, DW and FW, and how many plots
were involved.
L74 This is more than just mosaicking, that's photogrammetric restition, e.g. SfM-MVS photogrammetry
L83 "study" -> "studies"; which previous studies? provide references of studies that (apparently) failed to do so. This is a very narrow application domain; are
there such studies in other vegetation types, e.g. in studying forest biomass using UAV?
Section 2.3 - please briefly report the horizontal and vertical precision of each orthomosaic and DSM. In Section 2.4.2, the precision of the DSM difference could be
estimated by adding up the variances (assuming their independence between DSMs), i.e. precision_diff = sqrt(precision_1^2 + precision_2^2).
Eq. (5): "VIs" is a plural, there are 12 VIs, therefore there should be 12 different alphas and this becomes a multiple linear regression model. The authors should
correct this equation and state more clearly what they did, how they etimated the model coefficients. Usually alpha denotes the intercept (the constant), beta_i are
the coefficients of the individual predictor variables. (P.S. Later I realized that only simple linear regressions were considered, which is not at all clear at this
point.)
Eq. (4) and (5) - I don't understand why dry weight should relate linearly to CV while FW only after accounting for / multiplying with VIs. What is the underlying
assumption? It would probably become clearer if the models were reexpressed as
density^ = alpha + beta_1 VI_1 + ... + beta_p VI_p
(but note that these alpha and beta values are different from the ones obtained with the authors' models, because I did not just divide their equations by CV.)
For dry weight, the authors basically propose using a fixed 'apparent' density (or conversion factor).
L296 K-S-test: If the P-value is <0.001, the authors must reject the null hypothesis of having the distribution hypothesized under the null hypothesis. I.e. they
must reject normality. However, I think this is of very little relevance. What is normality needed for, in the rest of the paper? In Fig. 3, spread around the
regression line doesn't seem to be completely skewed, and there are no outliers, so everythin seems fine. With large sample sizes, even small, irrelevant departures
from normality will lead to a rejection of normality.
L325-330 Move this to the Methods section. Avoid repetition of details of methods in the results section.
L332-340 All text introducing accuracy measures should be moved to Methods section.
L345 I think this is the first time the authors mention that linear regressions using VIs as predictors are used as a benchmark for the proposed CV-based model. Yet
important information is missing (see earlier comments regarding Eq. (4) and (5)).
L417ff - I'm not if I agree. The authors' assumption is that the proposed method requires no further calibration. As can see, the model has two coefficients, alpha
and beta, which were empirically estimated from this sample. How can the authors be so sure that they need not be re-estimated next year? This would require new
ground truth data, i.e. destructive sampling. Also, (how) can the authors be so sure that the bias in the crop height model stays the same?
L467ff - The authors emphasize the genotype-dependence of VI-based approaches, however they do not demonstrate that their own approach is not genotype dependent.
Considering the data available in this study, the authors have an excellent opportunity to examine a possible genotype-dependence of their proposed method, or to
empirically demonstrate that it works equally well for different genotypes.
L494 A limitation of the comparison presented here is that only simple linear regressions were considered, although multiple linear regression (or more advanced
regression techniques) would be a quite obviously more promising candidate method. One study that comes to my mind (but it is not crop-related, it's from a shrub
environment) is Zandler et al. (2015) in Remote Sensing of Environment, who apply penalized linear regression techniques and VIs to estimate above-ground biomass.
Minor editorial changes:
L16 "does" -> "do" (plural/singular mismatch)
L22-23 SEE: which unit?; K: if this is kappa, use the Greek letter
L24 and L438 "metrices" - metrics? This can be omitted.
L69 "using ... identifying" - omit "identifying". In (iii), a word is missing, insert "from" or "such as"?
L82 A word is missing, e.g. "information"
L103 "a fusion based image analysis approach was used" - This seems to a be wordy version of just saying that the results of CGM, CC and CV models were combined?
L140 "plat" -> "plant"
L240 "separate" -> "separates"
L250 "fusion" -> "combination"
Eq. (1): CHM and CC should both have indexes i and j, not the absolute value of their differences, and the sum should be a double sum, one running from i=1 through m
and the other from from j=1 to n. Note that this is an (m+1) x (n+1) matrix if the summation start with index 0. On the other hand, is this equation really
necessary?
Eq. (2) - not really necessary
Equations should not be listed but inserted in the text
Eq. (4) and (5) the dot is a multiplication? use the appropriately centered dot instead, i.e. \cdot in LaTeX or suitable equivalent in Word
Use italics for mathematical symbols consistently even in the text.
L284 is not very informative / relevant. I think this paragraph can be shortened to basically only mention the problem-specific software, i.e. Pix4D and some of the
not-so-obvious Python packages.
L300, L366, L384: just write "p-value <0.001"
L304 The intercept of the linear model in Fig. 3 is not the mean difference between CHM-derived and ground truth plant height. In this sentence it would be more
appropriate to report the mean difference of ground truth height minus crop height. The mean difference seems to be close to 20 cm, according to Fig. 3.
L310 "the advantages of" - this sounds like the authors are biased towards their model; just omit these words
L311-312 Move this information to the figure caption of figure 4. The figure caption does not state at all what kind of information is displayed.
Table 2 - not only shows accuracy, also confusion matrix. But units of integer values are missing. Figure caption should mention that these are numbers of points or
pixels (not e.g. square meters).
L479 "multi-growth stages" -> "multiple growth stages"
L493 "singular valid" -> "single"
L499-500 "advanced analytics" - re-word; at least the statistical data analysis part was not advanced analytics.
Author Response
Uploaded response in word file
